# Mafosfamide Boosts GMI-HBVac against HBV via Treg Depletion in HBV-Infected Mice

**DOI:** 10.3390/vaccines11061026

**Published:** 2023-05-25

**Authors:** Qin Lin, Yiwei Zhong, Bin Wang

**Affiliations:** 1Key Laboratory of Medical Molecular Virology (MOE/NHC/CAMS), Shanghai Frontiers Science Center of Pathogenic Microorganisms and Infection, School of Basic Medical Sciences, Fudan University, Shanghai 200032, China; 2Shanghai Institute of Infectious Disease and Biosecurity, Fudan University, Shanghai 200032, China; 3National Clinical Research Center for Aging and Medicine, Huashan Hospital, Fudan University, Shanghai 200040, China; 4Children’s Hospital of Fudan University, Shanghai 201102, China

**Keywords:** chronic hepatitis B infection, therapeutic vaccines, regulatory T cells, functional cure, GMI-HBVac, mafosfamide

## Abstract

Chronic hepatitis B infection remains a significant worldwide health burden, placing persons at risk for hepatocellular cancer and hepatic fibrosis. Chronic hepatitis B virus (CHB) infection is characterized by elevated levels of immunosuppressive regulatory T cells (Tregs), which can inhibit the function of effector T cells and lead to an insufficient immune clearance response against HBV. Theoretically, suppression of Treg cell functionality and percentage could increase anti-HBV reactivity in CHB-infected patients, although this has not yet been explored. We attempted to enhance our previously established anti-CHB protocol utilizing the GM-CSF+IFN-α+rHBVvac regimen (GMI-HBVac) by incorporating mafosfamide (MAF), which has been utilized in anticancer therapy in the past. Intravenous administration of MAF to rAAV8-1.3HBV-infected mice resulted in a dose-dependent reduction of Tregs in the blood, rebounding to pretreatment levels 10 days later. To assess the potential benefit of adding MAF to the anti-CHB protocol, 2 μg/mL MAF was combined with the GMI-HBVac as an anti-Treg treatment in an HBV-infected animal model. When rAAV8-1.3HBV-infected mice were immunized with MAF+GMI-HBVac, peripheral blood Tregs decreased significantly, leading to dendritic cell activation, HBV-specific T cell proliferation, and the upregulation of IFN-gamma-producing CD8^+^T cells. In addition, MAF+GMI-HBVac vaccination stimulated T cell infiltration in HBV-infected livers. These effects may contribute to an enhanced immune response and the clearance of HBV-associated antigens, including serum HBsAg, serum HBcAg, and HBcAg^+^ hepatocytes. Overall, this is the first indication that MAF can act as an adjuvant with GMI-HBVac to deplete Tregs in mice with an established CHB infection. This unique therapeutic vaccine regimen produced a functional cure, as revealed by the remarkable clearance of HBsAg.

## 1. Introduction

Hepatitis B is a potentially fatal liver disease caused by infection with the hepatitis B virus (HBV), and it has become a major global health issue. HBV is an enveloped, hepatotropic, non-cytopathic DNA virus that can cause acute and chronic hepatitis [1]. Chronic HBV infection (CHB) puts people at risk of cirrhosis, hepatocellular carcinoma, and death in the advanced stage of the disease. In 2019, the all-age prevalence of chronic HBV infection was about 4.1%, corresponding to 316 million infected people worldwide, and approximately 555,000 people die each year from HBV-related diseases [2]. In Asia, the estimated percentage of adults with chronic HBV infection ranges between 3% and 7.9%, posing a serious public health concern. Finding a cure for chronic hepatitis B remains a formidable challenge [3].

The World Health Organization (WHO) issued recommendations in 1992 and 2009 urging all nations to include hepatitis B vaccination at birth in their national immunization schedules and to use oral antivirals to suppress HBV infection and slow the progression of the disease [4]. In 2016, the WHO adopted the Global Health Sector Strategy (GHSS) regarding viral hepatitis with the goal of eradicating the disease by the year 2030. The GHSS document outlined the necessary steps, which included the following five core intervention areas: (1) HBV vaccination; (2) prevention of mother-to-child transmission of HBV; (3) safe blood and injection procedures; (4) harm reduction; and (5) testing and treatment of HBV and HCV [5]. In 2022, the 2022–2030 GHSS viral hepatitis goal was approved. The strategies aim to end viral hepatitis epidemics as part of advancing universal health coverage, primary health care, and health security in a world where everyone has access to high-quality, evidence-based, people-centered health services and can live healthy and productive lives [6]. In 2016, only 12 countries were on track to meet hepatitis elimination targets, with 87% of infants receiving three doses of HBV vaccination in their first year and 46% receiving a timely dose of HBV vaccine at birth. The WHO guidelines recommend a single quality-assured serological in vitro diagnostic (IVD) test for HBV infection, but in 2018, due to a lack of financial support, the number of HBV infections diagnosed serologically remained at 9% [7]. Community-based and self-administered integrated testing services enable immediate treatment, and a simulation study found that since vaccination efforts have increased, new cases of hepatitis B have been avoided, but liver disease mortality is expected to rise under current testing and treatment interventions [8].

Current therapies, with interferon and nucleos(t)ide analogs (such as Tenofovir, Entecavir, Emtricitabine, Telbivudine, Lamivudine, and Adefovir), are recommended by WHO as the preferred first-line antiviral treatment. They can prevent the development of cirrhosis and hepatocellular carcinoma, but do not eradicate the virus, and only rarely clear HBsAg. Both approaches yield low rates of HBsAg seroclearance, and there is a viral rebound if the drugs are withdrawn. The persistence of cccDNA and integrated DNA remain critical challenges for these antiviral approaches, and long-term medication can impose an enormous burden on patients, as well as incur high costs. New antiviral and immunomodulatory therapies are being tested in clinical trials, with the goal of achieving a functional cure (i.e., clearance of HBsAg). Many new drugs have failed to meet their primary efficacy targets when tested as monotherapies, prompting the investigation of various combination treatment options. Combination therapies are likely to be necessary for a full clinical cure for hepatitis B. The consensus view is that antiviral drugs should be combined with an effective immunotherapeutic treatment.

Several HBV components that are produced during infection potently induce regulatory T cells (Tregs, defined as CD4^+^CD25^+^Foxp3^+^) that suppress T cell proliferation and effector functions. CHB has been linked to a significant increase in the proportion of Tregs [9,10,11,12]; immunosuppression and immunotolerance led by Tregs contribute to the failure of immunity to clear host cells infected with HBV and HBV-related antigens (e.g., HBsAg, HBeAg, HBV DNA) and underlies chronic infection. In an analogous situation in murine tumor models, Tregs have also been shown to impede the development of antitumor immune responses. Removing CD4^+^CD25^+^Foxp3^+^ Tregs boosted immune responses against tumor antigens after immunotherapeutic agents were administered. Similar evidence was found in chronic HBV infection [13,14]. Accordingly, methods of impairing Treg function may hold the key to successful immunotherapeutic treatment for chronic HBV infection. There would be significant therapeutic potential in delivering an immunotherapeutic agent to suppress the function of Tregs, exhibiting minimal side effects on normal immune cells and achieving the goal of a functional cure for CHB infection. However, most immunotherapeutic approaches have had difficulty circumventing or overcoming immunosuppression and immunotolerance [15,16].

Cyclophosphamide (CTX) and mafosfamide (MAF) are two alkylating agents that are used in cancer chemotherapy to suppress Treg function [17,18,19]. Immunosuppression of cell proliferation and excessive immune response, i.e., to prevent organ rejection in transplantation, necessitates high doses of the agent. Nonetheless, low doses of the agent can boost immune responses to pathogenic antigens by depleting Tregs and unidentified immunosuppressive cells, termed MDSCs. CTX is hydrolyzed to phosphoramide mustard by the cytochrome P450 system (phosphoramidase/P450-enzyme/phosphatase) in the liver. Phosphoramide mustard has the ability to alkylate DNA at the N7 position of guanine, resulting in DNA inter-strand crosslinks and, as a result, cell apoptosis. Although MAF is a derivative of CTX, it does not require hydrolysis in the liver to function, making it safer than CTX [20,21]. It has recently been demonstrated that low doses of CTX or MAF can reduce the percentages of CD4^+^CD25^+^Foxp3^+^ Tregs and directly attenuate Treg suppression. Treatment with CTX or MAF also promotes the apoptosis of other immunosuppressive cells. In other approaches, several studies have found that GM-CSF can stimulate the activation of phagocytes and dendritic cells [22,23,24]. Interferon-α can induce specific degradation of HBV covalently closed circular DNA (cccDNA) in HBV-infected cells via up-regulated APOBEC3A and APOBEC3B cytidine deaminases, without causing hepatotoxicity [25,26]. Furthermore, previous research in our laboratory has shown that cytokines such as GM-CSF and IFN-α can effectively boost the immune response induced by the hepatitis B vaccine [27,28]. All of this emphasizes the possibility of curing CHB infection.

We investigated the impairment of immune suppression induced by Tregs with MAF or CTX in order to potentiate the immunogenicity of the GMI-HBVac when MAF or CTX was used as an adjuvant. We tested for enhancement of the efficacy of the immunotherapeutic treatment of mice with CHB caused by rAAV8-1.3HBV infection. To the best of our knowledge, the therapeutic effect of MAF or CTX on Treg in CHB infection has not previously been investigated. This approach could be modified and lead to a functional cure for CHB infection in human clinical trials.

## 2. Materials and Methods

### 2.1. Animals, Viruses, and Reagents

Animals: Six- to eight-week-old male C57BL/6 mice of the wild-type strain were acquired from Silaike Laboratory Animal Co., Ltd. (Shanghai, China). All mice were housed in separate, ventilated, pathogen-free cages following the animal care guidelines of Fudan University.

Viruses: The rAAV8-1.3HBV has a 1.3-fold HBV genome (genotype D, serotype ayw) packed into a recombinant AAV8 vector. The rAAV8-1.3HBV was acquired from Paizhen Biotechnology Co., Ltd. (Guangzhou, China).

Reagents: Recombinant Chinese hamster ovary cell-derived HBsAg (rHBVvac) and recombinant human GM-CSF were generously donated by Jingtan Biotech Corp. of China North Pharmaceutical Group (Shijiazhuang, China) and Huaqiao Hospital, respectively (Putian, China). Human recombinant Interferon-α2b (IFN-α2b, Beijing Kawin Technology Share-Holding Co., Ltd., China) was purchased from Ruijin Hospital’s Pharmacy (Shanghai, China), and purified recombinant HBsAg protein (rHBsAg) was purchased from Guikang Biotechnology, Ltd. (Shanghai, China). Both the HBsAg-specific CTL epitope S_208–215_ (ILSPFLPL; H-2b-restricted) and the Ovalbumin (OVA)-specific CTL peptide OVA_257–264_ (SIINFEKL; H-2b-restricted) were manufactured by the Genscript Biotech Corporation (Nanjing, China). CFSE was purchased from Invitrogen, and the antibodies used in flow cytometry and immunohistology analysis and Brefeldin A (BFA) were obtained from Becton, Dickinson & Co., BioLegend Inc., and eBioscience, respectively. Fetal bovine serum (FBS, Gibco) was purchased from Life Technologies Corporation (Shanghai, China). Ionomycin and phorbol myristate acetate (PMA) were acquired from MedChemExpress company. Cyclophosphamide (CTX, purity ≥ 98%) and mafosfamide (MAF, purity ≥ 95%) were obtained from Sigma-Aldrich and Santa Cruz Biotechnology. EasySep™ Mouse T Cell Isolation Kits were used (Stemcell, #19753 for CD8, #19752 for CD4). FACS antibodies for monocytes: CD11b (clone: M1/70, BioLegend), CD11c (clone: HL3, BD), CD80 (clone: 1610A1, eBioscience), CD86 (clone: GL-1, BioLegend), MHC-I (clone: AF6.88.5.5.3, eBioscience), MHC-II (clone: M5/114.15.2, eBioscience), Ly6G (clone: RB6-8C5, eBioscience), and Ly6C (clone: HK1.4, BioLegend). For T cells: CD4 (RM4-5, BioLegend clone) and CD8 (clone: 53-6.7, eBioscience). Intracellular cytokine: anti-IL4 (clone: 11B11, BioLegend) and anti-IFN-γ (clone: XMG1.2, BioLegend).

### 2.2. CHB Animal Model

Each male mouse was injected via the tail vein with 1 × 10^10^ TCID50 of the rAAV8-1.3HBV, following the manufacturer’s instructions. All mice were bred under controlled environmental conditions. Serum from infected mice was analyzed to determine the levels of HBV DNA, HBsAg, anti-HBsAg, and HBeAg. All experimental procedures were performed under the guidance of the animal use and care committee.

### 2.3. Immunization

Mice infected with rAAV8-1.3HBV were randomly divided into six groups (*n* = 6), as shown in Table 1. Subcutaneous injections of GM-CSF (10 μg), IFN-α (10,000 IU), and HBV vaccine (1 μg) were administered on day 0 following vaccination. On day 0, CTX or MAF was given intravenously, whereas mice in the control group did not receive CTX or MAF. Three subcutaneous injections of the same regimens were administered on days 14, 28, and 49. After the injections, the mice were sacrificed at intervals, and spleen and peripheral blood cells were removed and examined.

### 2.4. Delayed-Type Hypersensitivity (DTH) Assay

Seven days after the third and fourth immunizations, 10 μg of rHBsAg was injected into the left footpad, and phosphate-buffered saline (PBS) was injected into the right footpad as a control. After 24 h and 48 h, the swelling was measured with a vernier caliper. The results are reported as the mean standard deviation of the thickness of the left and right footpads.

### 2.5. Isolation of Splenocytes

Mouse spleens were crushed in RPMI1640 to release the cells. The cells were centrifuged at 400–600× *g* for 5 min at 4 °C, the supernatant was discarded, and the cells were resuspended in 2 mL pre-cooled 1× red blood cell lysis buffer on ice for 5 min. The cell suspensions were washed with 10 mL ice-cold PBS and centrifuged at 400–600× *g* for 5 min at 4 °C. After discarding the supernatant, the cells were resuspended in RPMI1640 and adjusted to 2–3 × 10^6^ cells/mL.

### 2.6. Isolation of T Cells (CD8^+^ T Cells, CD4^+^ T Cells)

The cells were resuspended in 2 mL RPMI1640 at a concentration of 1 × 10^8^ cells/mL, and normal rat serum was added at a concentration of 50 μL/mL of cells. The EasySep™ Mouse T Cell (CD8^+^ or CD4^+^) Isolation Cocktail was added, and the sample was incubated at room temperature for 10 min. The EasySep™ Streptavidin RapidSpheres™ was added at 125 μL/mL of cells and incubated for 5 min at room temperature. RPMI1640 was added to the cell suspension to a total volume of 2.5 mL and mixed by pipetting up and down three times. The tube was placed in a magnetic cylinder for 2.5 min at room temperature. The separated (free) cells were then poured into a new tube. Negative selection could then be conducted on the magnetically labeled cells that remained bound inside the original tube.

### 2.7. Cell Culture for Intracellular Cytokine Analysis

After sacrifice, the mice were disinfected with 75% alcohol for 10 min. The spleens were removed aseptically. Each spleen was placed in a sterile Ep tube with RPMI1640 full culture medium, crushed with a sterile syringe plunger, washed with PBS, and filtered through a cell strainer (scorch-sterilized copper mesh) to remove large tissue pieces. The filtered cell suspension was centrifuged at 1500 rpm for 3 min. The supernatant was discarded, and 600 μL of red blood cell lysis buffer was added to lyse the red blood cells for 1 min. RPMI1640 complete medium (600 μL) was added to stop the lysis. After centrifugation at 1500 rpm for 3 min, the cells were resuspended in RPMI1640 complete medium and adjusted to 1 × 10^7^ cells/mL; then, 100 μL of cell suspension was added to the wells of a 96-well cell culture plate (1 × 10^6^ cells/well) and stimulated with HBsAg-specific CTL epitope S208-215 (ILSPFLPL, H-2b-restricted). A negative control was stimulated with OVA257-264 (SIINFEKL) and a positive control with PMA and ionomycin. The cells were incubated at 37 °C in a cell culture incubator with 5% CO_2_ for 24 h. Brefeldin A was added for the last 6 h to block cytokine secretion. BFA-blocked cells were collected and washed for examination.

### 2.8. In Vitro T Cell Proliferation Assay

On day 14 following the fourth immunization, all animals were sacrificed for the preparation of single-cell suspensions of splenocytes. To assess splenocyte proliferation, a commercially available cell proliferation dye (CFSE, Invitrogen) was utilized. As cells divide, the fluorescence intensity of primordial cells diminishes. Splenocytes (5 × 10^5^ cells) were incubated with CFSE at a concentration of 1 μM. The cells were incubated at 37 °C in the dark for 10 min before being resuspended in RPMI1640 with 10% FBS. After washing, the cells were moved to plates and re-incubated in the presence of anti-CD28 (100 ng/mL) with HBsAg-specific CTL epitope S_208–215_ (1 μg/mL) as a specific stimulator. Positive control cells were re-incubated with anti-CD3 (1 μg/mL) and anti-CD28 (100 ng/mL), while negative control cells were re-incubated with OVA-derived peptide OVA_257–264_ (1 μg/mL). All plates were incubated for 72 h at 37 °C in a humidified environment containing 5% CO_2_ before flow cytometry (FCM) analysis was performed.

### 2.9. Serological and Biochemical Analysis

Seven days after the last vaccination, serum HBsAg, HBsAb, and HBeAg levels were measured using ELISA kits acquired from Kehua Bioengineering Co., Ltd. (Shanghai, China). Serum ALT activity was determined using an ALT kit from BioSino Bio-technology and Science Inc. (Beijing, China).

### 2.10. HBV DNA Quantitation

Real-time quantitative PCR was used to quantify serum HBV DNA using a kit from Kehua Bio-engineering Co., Ltd. (Shanghai, China). The minimum detectable concentration was 100 IU/mL.

### 2.11. In Vivo Cytotoxic Lysis Assay

Splenocytes from naive C57BL/6 donor mice were labeled with 15 μM CFSE and treated with 1 μg/mL S_208–215_ peptide to provide cells of high fluorescence intensity. An equal number of splenocytes were labeled with 1 μM CFSE and activated with 1 μg/mL of OVA_257–264_ peptide to provide cells of low fluorescence intensity. On day 14 after the last immunization, 2 × 10^7^ cells were adoptively transferred intravenously into immunized recipients using a 1:1 mixture of the two cell types. The recipients’ splenocytes were extracted after 8 h and assayed for FCM fluorescence intensity.

### 2.12. Histology and Immunohistochemistry (IHC) Analysis

The liver and spleen were fixed overnight with 4% formalin and were then paraffin embedded. Hematoxylin-eosin (H&E)-stained 10-μm slices were examined by light microscopy to identify liver abnormalities. IHC analysis was used to investigate the infiltration of HBcAg- and HBsAg-positive hepatocytes and inflammatory cells. To assess the HBcAg, HBsAg, CD4, and CD8 markers, 10-μm liver or spleen slices were fixed with ice-cold methanol, dehydrated through graded alcohols, and subjected to antigen retrieval with 10 mM sodium citrate. Sections of the liver or spleen were treated with mouse monoclonal anti-HBcAg, anti-HBsAg, anti-CD4, and anti-CD8 antibodies. Sections were treated with goat anti-mouse horseradish peroxidase antibody for 1 h after being washed and incubated with TBST for 1 h. Finally, the slides were treated with 3,3-diaminobenzidine (DAB) chromogen solution, followed by hematoxylin counterstaining. Then, the integral OD of HBcAg, HBsAg, CD4, and CD8 staining was examined using Image-Pro Plus (Version 6.0, Media Cybernetics, Inc., Silver Spring, MD, United States of America) software.

### 2.13. Flow Cytometry (FCM)

Flow cytometry was performed with Diva software^TM^ (Version 8.0) on a flow cytometer (FACS LSR-Fortessa, Becton, Dickinson and Company, Shanghai, China). Single-cell suspensions were prepared and stained for 20 min at 4 °C on ice with specific monoclonal antibodies for cell surface markers (including CD3, CD4, or CD8). Unconjugated fluorescent antibodies were washed away with FACS buffer after incubation. The cells were fixed with 4% paraformaldehyde for 10 min and permeabilized with 0.1% saponin for 8 min to detect intracellular cytokines (IFN-r, IL-4, and IL-17A). Following fixation, the cells were washed with FACS buffer several times to remove the fixative. The appropriate fluorescently labeled anti-mouse monoclonal antibodies were then added and incubated on ice for 1 h. The cells were resuspended and analyzed with a BD Fortessa flow cytometer after a final wash. To identify nonspecific background staining, isotype-matched control mAbs were employed.

### 2.14. Statistics

GraphPad Prism Software 9.0 was used for statistical analysis (GraphPad, La Jolla, CA, USA). All results are displayed as means ± standard error of mean (SEM). For continuous variables, the Mann–Whitney U test was used to determine abnormal distribution, whereas the Student’s *t*-test was used to determine the normal distribution and homogeneity of variance. *p* < 0.05 was considered significant for all analyses.

## 3. Results

### 3.1. Production of rAAV8-1.3HBV-Infected Mice and Immune Tolerance Facilitated by Increased CD4^+^CD25^+^Foxp3^+^ Tregs

To examine the efficiency of our regimens, male C57BL/6 mice were injected intravenously with rAAV8-1.3HBV at 1 × 10^10^ TCID50 per mouse to produce a persistently infected mouse model for testing. To confirm rAAV8-1.3HBV infectivity in these mice, the HBV-related antigens were measured by ELISA, qPCR, and immunohistochemistry after 14 days and continuously monitored for half of the year (Figure 1A). The analysis revealed that HBsAg, HBeAg could be detected in serum by ELISA and qPCR after 2 weeks of infection, the levels of serum HBsAg and HBeAg were dramatically increased over baseline (cut-off value), and the level of anti-HBsAg was below the baseline. Moreover, serum ALT was also increased over the baseline (Figure 1D). H&E staining revealed mildly diffuse hepatocellular degeneration, together with swollen and vacuolated hepatocytes in HBV-infected mouse livers when compared to hepatocytes from healthy individuals. When immunohistochemistry and immunofluorescence staining were used to determine the expression of HBsAg and HBcAg in the livers of rAAV8-1.3HBV-infected mice, large amounts of HBsAg and HBcAg were found in infected hepatocytes, with widespread distribution (Figure 1C). Moreover, HBsAg and HBeAg persisted in the serum and hepatocytes for several months. These findings are consistent with the concept that clinical CHB individuals exhibit an active expression of HBV-related antigens without anti-HBsAg in serum. Thus, we verified that the persistently infected CHB-like mouse could be used for developing novel antiviral strategies for the functional cure of CHB infection.

The difficulty in treating chronic hepatitis B is the occurrence of immune suppression due to immune tolerance and excessive Tregs; many studies have reported that immune tolerance is associated with immunosuppressive cells, such as Tregs, induced by HBV infection. We found that Tregs of peripheral blood represented 7.13% of the total CD4^+^ T cell population in the CHB mice infected by rAAV8-1.3HBV, which was almost double the proportion in the negative control mice (4.25%; Figure 1B). This is consistent with the clinical observations that excessive production of Tregs impairs the immune response against HBV-related antigens, resulting in immune tolerance and chronic HBV infection.

### 3.2. Low Concentrations of Mafosfamide (a Cyclophosphamide) Contribute to Tregs Decrease, with Minimal Detrimental Effects In Vivo

High and low concentrations of CTX and MAF have opposite effects on immune function; accordingly, their active products after via dinstinguished metabolic pathways are also different. To deplete the excessive Tregs and break immune tolerance in CHB-infected animals, we first tested the pernicious effects of CTX or MAF on Tregs at different concentrations. CTX or MAF, with different doses, was administrated once by intravenous tail injection into five groups, as illustrated in Figure 2A. PBS was injected as a negative control. Doses for MAF were set at 0, 0.5, 1.0, 2.0, and 4.0 μg/mL, while CTX doses were set at 0, 15, 30, and 60 mg/kg. Peripheral blood mononuclear cells of the treated mice were collected on days 0, 2, 4, 6, 8, and 10 to determine the change of Tregs over time by FCM analysis.

As depicted in Figure 3A, we found that rAAV8-1.3HBV-infected mice treated with a low dose of MAF at 1 μg/mL had a striking decrease in the percentage of Tregs within 48 h, from 7.69% to 3.59%. Simultaneously, MAF treatment disturbed the balance of circulating lymphocytes, especially CD4^+^ and CD8^+^ T cells. In comparison, Tregs were more sensitive to MAF than the T cells and B cells. When the concentration of MAF was equal to or above 2 μg/mL (calculated based on a total of 2 mL of blood circulating in a mouse), the decrease in Tregs proportion was the greatest, from 6.72% to 0.97%, on the 4th day after the MAF administration. The effective depletion of Tregs was more than 80% and lasted for about 10 days. Tregs and T cells gradually returned towards the pretreatment levels after the MAF administration, suggesting a self-balancing mechanism of the immune system (Figure 2B and Figure 3A). When the concentration of MAF was above 4 μg/mL, the percentage depletion of Treg was slightly higher than with the 2 μg/mL treatment, but the loss of CD4^+^ T cells and CD8^+^ T cells reached 50%. In contrast, the loss of CD4^+^ and CD8^+^ T cells was less than 30% when mice were treated with MAF at 2 μg/mL. Interestingly, treatment with MAF caused slight weight loss, but the mice exhibited normal growth over time (Figure 3C,D). CTX was also examined for a depletion effect on Tregs. In contrast to MAF, the effect of CTX on Tregs at 15 mg/kg was weaker than that of MAF at 2 μg/mL, but more toxic on T cells than that of MAF at a similar dose level (Appendix A). Therefore, MAF at 2 μg/mL was investigated to determine if it could enhance the immunogenicity of GMI-HBVac.

### 3.3. Immunization of Mafosfamide Plus GMI-HBVac Breaks Immune Tolerance and Enhances the Function of DCs (CD11b^+^CD11c^+^) in rAAV8-1.3HBV-Infected Mice

Overcoming immune tolerance induced by CHB infection is essential in therapeutic vaccination. Immune tolerance is the main cause of immune anergy, while it is often attributed to APC (mainly DCs and Langerhans cells in this investigation), failing to recognize and present HBV-related antigens. APCs not only participate in the recognition and presentation of HBV-related antigens, but are also involved in the activation of T cells and B cells via the costimulatory molecules expressed on DC surface membranes, such as MHC I, MHC II, CD80, and CD86. To examine the adjuvanticity of MAF on DCs, rAAV8-1.3HBV-infected mice were immunized with MAF combined with GMI-HBVac. Peripheral blood was collected for FCM analysis on the third day after the fourth immunization.

The intensity of the DTH reaction is one of the indicators used to assess antiviral cellular immunity in vivo. Therefore, we determined the intensity of DTH to estimate the antigen (HBsAg)-specific cellular immune response induced by our regimen. DTH reaction (assessed by footpad swelling) was detected after rAAV8-1.3HBV infection when the mice were re-challenged by antigen (HBsAg) in the footpads. As depicted in the photos (Figure 4B), the swelling (assessed by footpad thickness) of rAAV8-1.3HBV-infected mice immunized with MAF+GMI-HBVac was greater than that of mice immunized with GMI-HBVac and the controls. The level of DTH was the highest 24 h after the re-challenge, and it gradually returned to the pretreatment level after 72 h, suggesting that robust anti-HBV antigen-specific cellular responses were induced with the MAF + GMI-HBVac in rAAV8-1.3HBV-infected mice. The DTH reaction induced by CTX + GMI-HBVac was slightly weaker than that of GMI-HBVac, but higher than that of mice treated with CTX alone. These results suggest that either CTX or MAF administration can indeed enhance the cellular immune responses induced by the GMI-HBVac immunization, but the effect associated with the MAF was superior to that of CTX. Moreover, the analysis of peripheral blood mononuclear cells (PBMC) demonstrated that the regimen of MAF plus GMI-HBVac could markedly contribute to the availability of immunogenic dendritic cells defined as CD11b^+^CD11c^+^. After immunization with MAF + GMI-HBVac, the rAAV8-1.3HBV-infected mice exhibited higher percentages (4.58%) of DCs compared with the mice administrated with MAF (1.84%) or PBS (0.84%), and this was greater than the 2.92% seen after GMI-HBVac and the 3.24% noted after CTX + GMI-HBVac administration (Figure 4C,D). The increase in DCs implied that the proliferation and maturation of DC were enhanced by the regimen. In addition to increased DCs, functional molecules, such as CD80 expressed on the membranes of DCs, were simultaneously upregulated (Figure 4E). This would facilitate the recognition of HBV-related antigens and its subsequent activation.

### 3.4. Mafosfamide Plus GMI-HBVac Induces Robust T-Cell Responses and Facilitates Cytokine Secretion in the Peripheral Immune System

Anti-HBV cellular immunity has been demonstrated as essential to clear persistent HBV infection [29,30,31,32,33]. To gain insight into this potent regimen, we further analyzed several parameters of cellular responses. Because several vital cytokines secreted by activated T cells have previously been shown to be indirectly involved in the clearance of HBV-infected cells [34,35,36,37,38,39], T cell cytokine profiles were investigated. Serum cytokine assays were performed by ELISA. When splenocytes from the persistently HBV-infected mice that had been treated with MAF plus GMI-HBVac were re-stimulated in vitro with 10 μg/mL HBsAg, the frequency of IFN-γ secreting CD8^+^ T cells was sharply increased to 9.25%. Those from infected mice immunized with CTX plus GMI-HBVac showed upregulated frequencies of IFN-γ secreting CD8^+^ T cells up to 6.07%, which was similar to the 6.78% of GMI-HBVac only immunized mice (Figure 5A). Additionally, the IFN-γ-producing CD4^+^ T cells also increased to 7.08% in HBV-infected mice that had been immunized with MAF + GMI-HBVac; this was more than the 3.9% frequency in mice immunized with GMI-HBVac and the 6% frequency in mice immunized with CTX + GMI-HBVac (Figure 5B,C). This demonstrated that T cells were remarkably enhanced in the peripheral blood, particularly in the function of CD8^+^ T cells. In addition, the level of IL-4 secreting CD4^+^ T cells was increased to 9.85%, which was higher than the 6.25% in mice immunized with GMI-HBVac and the 8.5% after immunization with CTX + GMI-HBVac (Figure 5D). IL-17-producing CD4^+^ T cells reached 1.51% after immunization with MAF + GMI-HBVac, compared with 1.45% after GMI-HBVac treatment and 1.45% after CTX + GMI-HBVac administration (Appendix A).

CD8^+^ T cell-mediated cytotoxic T lymphocytes (CTL) play a crucial role in specifically targeting and eliminating HBV-infected hepatocytes. To directly determine the CTL activity of CD8^+^ T cells induced by the regimen in vivo, we employed an in vivo CTL assay with antigen-pulsed splenocytes labeled with high or low concentrations of CFSE at either 15 μM or 1 μM as HBs antigen-specific or OVA antigen-specific (negative control) target cells, respectively. Eight hours after injection into the mice, cells with the different fluorescence intensities were identified by FCM. The selective loss of cells with high CFSE fluorescence was near 60% in the mice treated with MAF + GMI-HBVac, 35% for mice treated with CTX + GMI-HBV, and 41.9% for mice treated with GMI-HBVac. In contrast, the killing rates in control mice treated with MAF, CTX, and PBS were less than 10% (Figure 5E). These data indicated that the MAF + GMI-HBVac could robustly enhance the CTL response and should be more effective in the treatment of HBV-infected hepatocytes than the GMI-HBVac and CTX + GMI-HBVac regimens.

The antigen-specific proliferation of CD4^+^T cells and CD8^+^T cells is also important in restoring antiviral cellular immunity. We used CFSE staining to analyze the CD4^+^ and CD8^+^ T cell populations and their proliferative capacities after the immunizations. The splenocytes of mice were co-stimulated with anti-CD3 plus anti-CD28 in vitro for the positive control and stimulated with PBS for the negative control, and the splenocytes of the treated mice were stimulated with HBsAg peptide (S_208–215_). Immunization with MAF and GMI-HBVac triggered an expansion of activated/proliferating T cells that differed from those observed after the immunization with other regimens (Figure 5F,G). The proliferation ratio was 32% for CD4^+^ T cells and 40.8% for CD8^+^ T cells in the mice immunized with GMI-HBVac, whereas in mice immunized with MAF + GMI-HBVac, the proliferation ratio for CD4^+^ T cells was 41.7% and for CD8^+^ T cells, it was 56.2%. In comparison, when we immunized with the CTX + GMI-HBVac, we achieved an average proliferation of 39.1% for CD4^+^ T cells and 53.3% for CD8^+^ T cells. These data revealed that either CTX or MAF combined with the GMI-HBVac can indeed facilitate T cell activation and proliferation.

Spleen tissues were used for immunohistochemistry staining with specific anti-CD4 and CD8 antibodies. We observed that there were more enlarged and agglomerated cells in the CHB model mice after immunization with GMI-HBVac and MAF + GMI-HBVac, whereas the splenocytes of rAAV8-1.3HBV-infected mice treated with MAF were smaller and densely arranged. The shade of the brown spots marked by anti-CD8 was deepened, and the brown zone was larger and denser after the mice were immunized with MAF + GMI-HBVac. These phenotypes were more obvious than those seen in rAAV8-1.3HBV-infected mice immunized with other regimens. When the spleen was stained with anti-CD4 antibodies, the shade of brown spots marking the CD4^+^ T cells after immunization with MAF + GMI-HBVac was not different from that of the controls. Thus the proliferation of CD8^+^ T cells in the spleen had been selectively and specifically activated by HBV antigen (Appendix A).

These results demonstrated that better immunotherapeutic effects were achieved with MAF + GMI-HBVac compared with either CTX + GMI-HBVac or GMI-HBVac. There was significant enhancement of the levels of serum IFN-γ, IL-4, and IL-17; expansion of CD4^+^ and CD8^+^ T cells; and a robust CTL response mediated by the CD8^+^ T cells. Thus, the immune function was improved in the peripheral immune system when rAAV8-1.3HBV infected mice were immunized with MAF + GMI-HBVac.

### 3.5. Mafosfamide Plus GMI-HBVac Promotes the Infiltration of T Cells and Enhances IFN-γ Secretion in the Liver Microenvironment of rAAV8-1.3HBV-Infected Mice

Since the liver is the host organ of HBV infection and reproduction, the attraction and infiltration of activated CD8^+^ T cells into the liver provides the vanguard for the clearance of HBV infected hepatocytes. CHB infection shapes the particular liver microenvironment that leads to anti-HBV immune dysfunction. To further examine the specific immune responses against HBV infection in the liver after the treatments, we analyzed liver T cells, especially IFN-γ expressing CD8^+^ T cells, by immunohistochemistry assay and immunofluorescence staining. We used red fluorescence-labeled antibodies to stain IFN-γ^+^CD8^+^ T cells and green fluorescence-labeled antibodies to stain HBcAg^+^ hepatocytes (indicating HBV-infected hepatocytes) in the liver. As shown in Figure 6A,B, abundant activated CD4^+^ T cells and CD8^+^ T cells were crowded around the hepatocytes and liver vacuoles, with a trend of expansion. Notably, rAAV8-1.3HBV-infected mice immunized with MAF + GMI-HBVac harbored more infiltrated T cells, particularly CD4^+^ T cells, in the liver than the mice immunized with CTX + GMI-HBVac or GMI-HBVac. Consistent with the results of the immunohistochemical assay, data from immunofluorescence staining (Figure 7A,B) revealed that larger numbers of IFN-γ^+^CD8^+^ T cells were present around the HBcAg^+^ hepatocytes after immunizations with the MAF + GMI-HBVac than after immunization with either CTX + GMI-HBVac or GMI-HBVac. Particularly, IFN-γ^+^CD8^+^ T cells were observed in zones filled with highly expressing HBcAg^+^ hepatocytes (strong green fluorescent regions). This suggests that anti-HBV and cytolytic IFN-γ^+^CD8^+^ T cells were activated and recruited to HBV-infected hepatocytes after immunization. Additionally, the intensity of red fluorescence in these mice was higher than in those immunized with CTX+GMI-HBVac and GMI-HBVac. The implication is that the proportion of infiltrated T cells in the livers of mice immunized with MAF+GMI-HBVac was greater than that in mice immunized with CTX+GMI-HBVac and GMI-HBVac.

### 3.6. Mafosfamide Plus GMI-HBVac Vaccination Contributes to HBsAg Seroclearance and HBV-Infected Hepatocyte Clearance in Chronic HBV Infection

The degree of a therapeutic vaccine’s ability to clear hepatitis B antigens indicates the potential for clinical application, and the liver is the primary organ of HBV virus infection, as well as the primary site for therapeutic vaccine function. Therefore, to further assess the therapeutic effects of these regimens, we collected liver tissues of the rAAV8-1.3HBV-infected mice after the last immunization for analysis of HBsAg and HBcAg content. As depicted in Figure 7A,B, the HBcAg^+^ hepatocytes labeled with green fluorescence were greatly reduced in the livers of infected mice immunized with the MAF + GMI-HBVac compared with those immunized with GMI-HBVac or CTX + GMI-HBVac. This finding is consistent with the results of immunohistochemical analysis (Figure 8A,B), in which a greater eradication of HBcAg^+^ hepatocytes was observed after immunization with MAF + GMI-HBVac compared with other treatments, and the reduction of HBcAg^+^ hepatocytes in mice immunized with MAF + HBVac was greater than that in mice immunized with GMI-HBVac or CTX + GMI-HBVac. Thus, the HBV virus replication and de novo infection could have been suppressed by an enhancement of anti-HBV immune responses. Histopathologic examination of liver biopsy samples (Figure 8C) showed that cell infiltration, especially lymphocyte infiltration, was facilitated around the vacuoles in livers from rAAV8-1.3HBV-infected mice immunized with MAF + GMI-HBVac; the accumulations of infiltrated cells were denser and affected most portal tracts. Concurrently, the vacuolated and swelled hepatocytes were obviously decreased, with small and few vacuoles. Few necroinflammatory and necrotic lesions were found after immunization with MAF + GMI-HBVac, which indicated that the regimen of MAF + GMI-HBVac induced a specific anti-HBV immune response with the remission of liver injury.

Serological markers of chronic hepatitis B infection in the patient’s blood are the key to determining the infection status; an essential hallmark of functional cure in CHB patients is that the level of HBsAg becomes negative, and the anti-HBsAg antibody becomes positive. We collected the blood of the rAAV8-1.3HBV-infected mice after each immunization for analysis using ELISA kits to determine the HBsAg, HBeAg, anti-HBsAg, and Q-PCR for HBV DNA, and also conducted a colorimetric test for ALT. As shown in Figure 9, we found that the serum HBsAg of the infected mice was substantially reduced after the second immunization with MAF + GMI-HBVac, and it continued to decline with increasing immunizations. The decrease was significantly greater than that achieved by the other treatments (Figure 9A). There was only a slight drop of serum HBeAg in the HBV-infected mice immunized with MAF + GMI-HBVac compared with the controls (Figure 9B). Most importantly, HBsAb, as the main force to neutralize HBsAg, was detected in the rAAV8-1.3HBV infected mice after the second immunization with MAF + GMI-HBVac, which continued to increase along with the number of immunizations, and it was at its greatest level after the 4th immunization. The HBsAb remained at a higher level for 24 weeks in mice immunized with the MAF + GMI-HBVac than in the mice immunized with other regimens (Figure 9C). During the treatment, there was a slight upward trend in serum alanine aminotransferase (ALT) levels, and there was no significant difference among these regimens (Figure 9D). No rebounds were found in either HBsAg or HBcAg after immunization with the MAF + GMI-HBVac.

In conclusion, the examination of peripheral blood and liver tissue directly confirmed that immunization of CHB-infected mice with MAF + GMI-HBVac can effectively lead to a breaking of immune tolerance and a rebuilding immune function to enhance anti-HBV responses. The new therapeutic strategy could not only increase intrahepatic IFN-γ-producing CD8^+^ T cells and induce HBsAg-specific antibodies, but it could also clear HBV-related antigens and suppress the replication of HBV infection and the production of HBcAg and HBeAg.

## 4. Discussion

Immunocompromised individuals who become infected with HBV are more likely to develop chronic HBV infections and immune tolerance to HBV-related antigens. The main barriers that prevent the efficacy of CHB treatments are systemic immune tolerance and exhaustion, both of which are caused by chronic HBV infection. The mechanism by which the host immune system develops immune tolerance to HBV is unknown, but previous research showed that the numbers and proportions of immunosuppressive cells, particularly Tregs and myeloid-derived suppressor cells, are significantly increased in patients with chronic hepatitis B. This increase in immunosuppressive cells impedes the anti-HBV responses, notably by reducing the availability of the specific CD8^+^ T cells that produce IFN-γ. These are the most important lymphocytes for anti-HBV immunity. To properly treat chronic hepatitis B infection, immunosuppressive cells must be decreased to restore the protective functions of the immunocytes. The depletion of Tregs is the most efficient method for abrogating immunosuppressive and immunological tolerance with an appropriate immunomodulator, and it may lead to a functional cure for CHB infection [9,16,40,41].

Current treatments with drugs containing nucleosides or nucleotides, in conjunction with interferon-α, result in delayed and/or decreased rates of liver failure, HCC, and associated comorbidities. To avoid HBV reactivation and hepatic flares, however, their use is not prolonged, and their effects are not sustained. The large size of the patient population necessitates the development of novel therapeutic techniques for HBV infection that result in a sustained decrease in off-treatment HBsAg. The distinct liver microenvironment created by persistent infection and changes in HBV-specific immunity must be addressed by innovative methods.

In our investigation, MAF and CTX were chosen as adjuvants or immunomodulators for the HBV therapeutic vaccine to drive Tregs to undergo programmed death via DNA crosslinking and apoptosis, thereby reducing the degree of immunosuppression caused by the virus. The enhanced therapeutic efficacy of vaccination that we observed when MAF treatment was added to GMI-HBVvac was consistent with the prediction of synergism, and it may exhibit wider clinical relevance. Verification that the mechanism entailed decreased levels of Treg cells was obtained. The finding that Tregs were more sensitive to MAF than the CD4^+^ or CD8^+^ cells (Figure 3) would account for the enhanced immune responses to the vaccine when MAF was added, seen in the activation of dendritic cells (Figure 4), increased cellular cytotoxicity (Figure 5), elevated antibodies against HBV antigens, and decreased HBV antigen levels (Figure 9). The enhanced recruitment of specifically activated CD4^+^ T cells and IFN-γ-producing CD8^+^ T cells around the HBV-infected hepatocytes, as well as and enhanced clearance of antigen from the liver (Figure 6, Figure 7 and Figure 8), were consistent with functional cure using the MAF + GMI-HBVac regimen, a key clinical objective. These immune responses and HBsAg clearance suggested that MAF + GMI-HBVac outperformed the HepTcell vaccine (Peptide+IC31, Clinical Phase II), GS4774 (HBsAg + HBcAg + HBx, Clinical Phase II), and the TG-1050 vaccine (Ad5 delivery, Clinical Phase I) [42,43,44,45]. In comparison to the Nasvac therapeutic vaccine (HBsAg+HBcAg, Clinical Phase III), our regimen required fewer doses and elicited a stronger antibody response [46]. These findings indicate that the MAF + GMI-HBVac regimen may provide the possibility for a functional cure for CHB individuals in the future.

## Figures and Tables

**Figure 1 vaccines-11-01026-f001:**
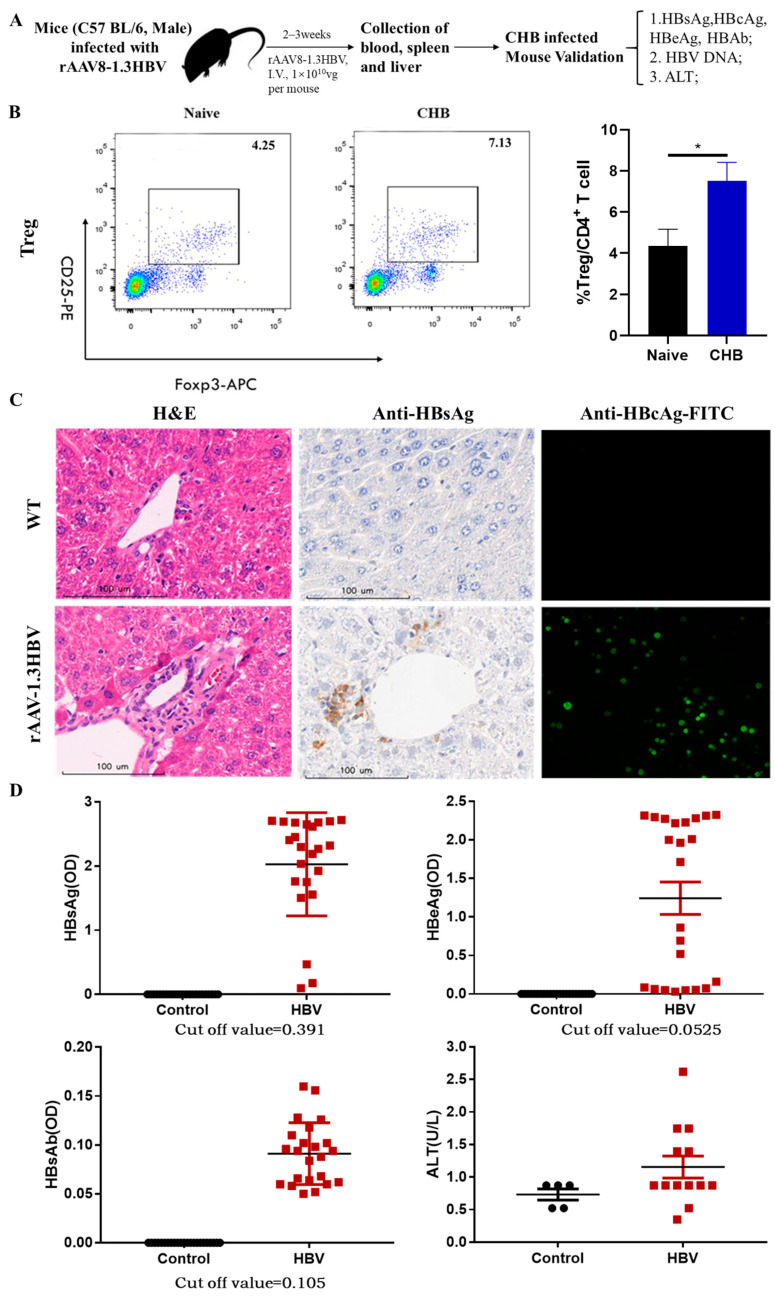
Production and verification of mice infected with rAAV8-1.3HBV. (**A**) Production and confirmation of CHB mice using the rAAV8-1.3HBV model. (**B**) Levels of CD4^+^CD25^+^Foxp3^+^ Tregs from the peripheral blood of mice infected with rAAV8-1.3HBV were compared to those of healthy mice. (**C**) Liver sections were stained with H&E, anti-HBsAg, and anti-HBcAg-FITC. (**D**) Serum levels of HBsAg, HBeAg, and ALT were evaluated in mice that had been infected with rAAV8-1.3HBV and in healthy controls using ELISA. * (*p* < 0.05) indicates significant difference.

**Figure 2 vaccines-11-01026-f002:**
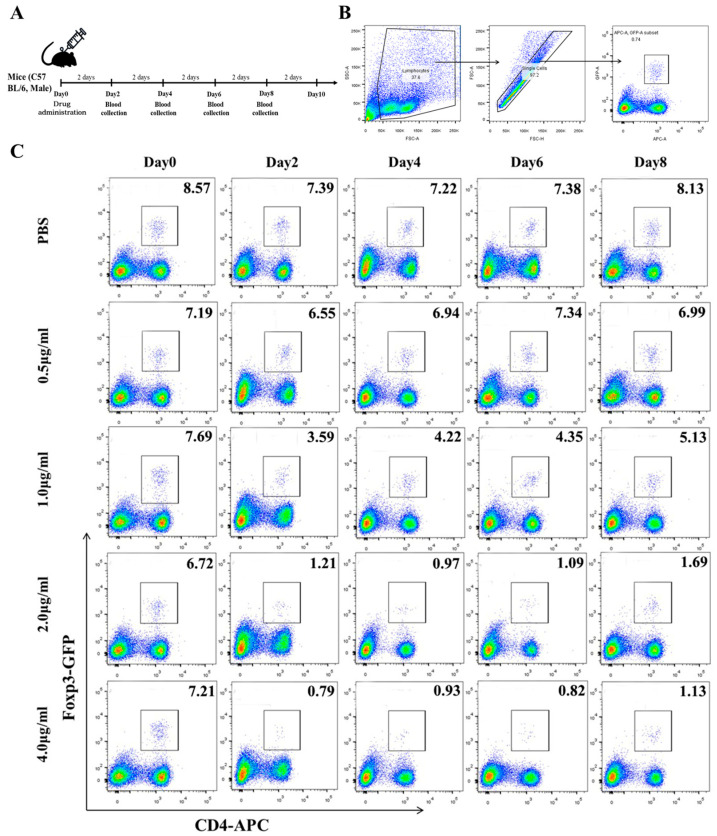
Determining the optimal MAF concentration for use as an adjuvant. The regimens were administered according to scheme (**A**), and blood samples were obtained at regular intervals (every two days). (**B**) Gating strategy for Treg. (**C**) The data collected from the FCM were analyzed using FlowJo-V10.

**Figure 3 vaccines-11-01026-f003:**
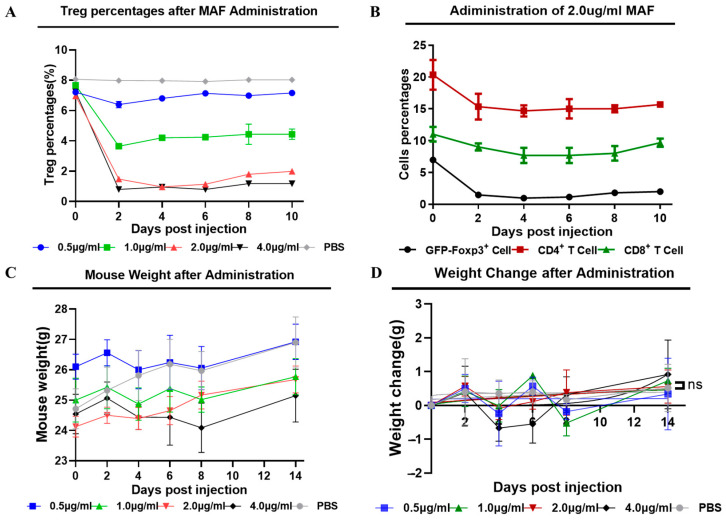
Determining the optimal MAF concentration for use as an adjuvant. (**A**) The percentage of Tregs present before and after the injection of MAF at each dose. (**B**) Treg percentage change following the injection of MAF at 2 μg/mL. (**C**) The general trend of mouse weight after MAF treatment. (**D**) The difference in weight before and after the administration of MAF.

**Figure 4 vaccines-11-01026-f004:**
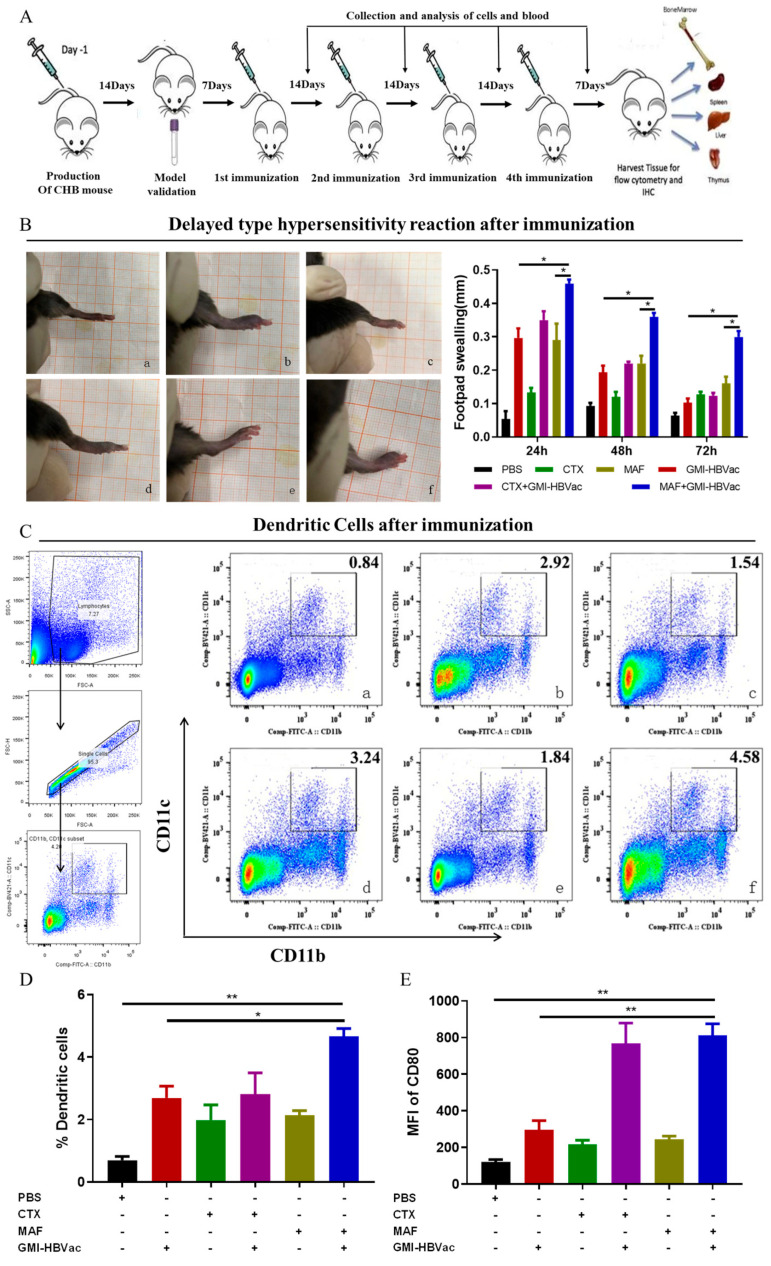
DCs and DTH responses were measured post-immunization. (**A**) A flowchart of the vaccination process, along with the associated regimen. (**B**) Mouse footpad swelling was assessed using a vernier caliper. (**C**) DCs from the peripheral blood were examined using FCM with a gating strategy. (**D**) Levels of DCs from the peripheral blood were examined using FCM. (**E**) The level of costimulatory molecules expressed on DCs was determined: ((**a**) PBS; (**b**) GMI-HBVac; (**c**): CTX; (**d**) CTX + GMI-HBVac; (**e**) MAF; (**f**) MAF + GMI-HBVac). * (*p* < 0.05) indicates significant difference, ** (*p* < 0.01) indicates highly significant difference.

**Figure 5 vaccines-11-01026-f005:**
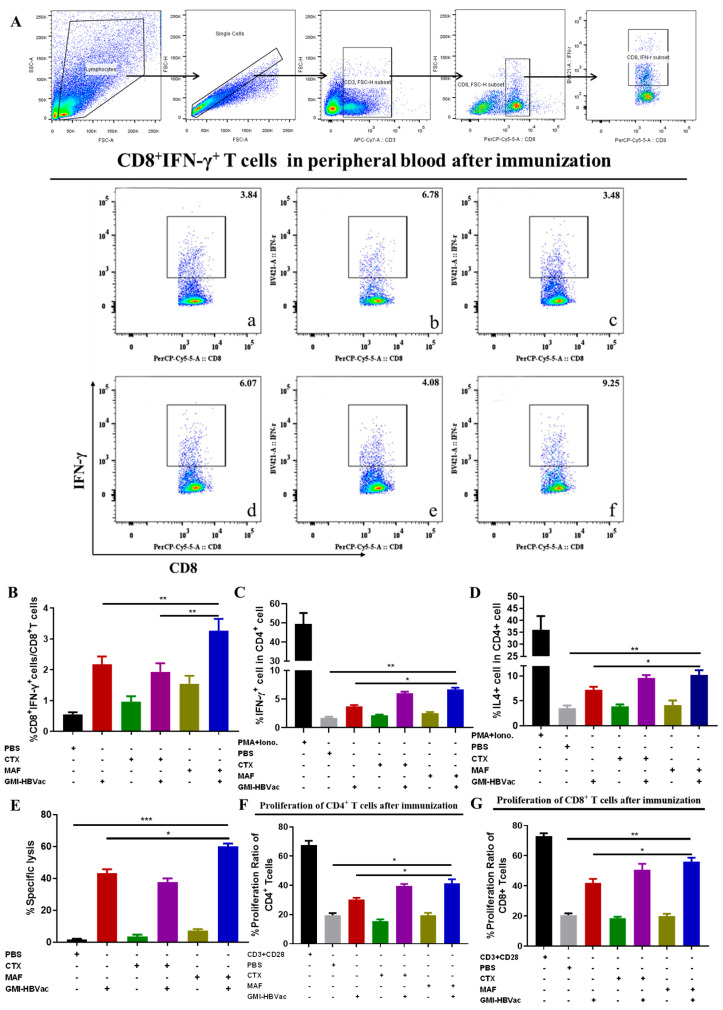
The immunization of rAAV8-1.3HBV-infected mice with MAF + GMI-HBVac enhances the immune response in the periphery. (**A**,**B**) IFN-γ^+^CD8^+^ T cells were examined by FCM after vaccination. (**C**) IFN-γ^+^CD4^+^ T cells following vaccination. (**D**) IL-4^+^CD4^+^ T cells following vaccination. (**E**) The functionality of CTL following immunization. Target cells were isolated from the spleen of naive mice, pulsed in vitro with an HBsAg peptide (S_208–215_: ILSPFLPL) or an OVA peptide (OVA_257–264_: SIINFEKL), and then labeled with a high concentration of CFSE for the cells that had been pulsed by HBsAg peptides and a low concentration of CFSE for the cells that had been pulsed with OVA. These two types of marked cells were combined in an equal ratio before being intravenously adopted by immunocompromised individuals. After an incubation period of 8 h to allow for specific cytolytic reactions, the reduction of CSFE was measured by flow cytometric analysis. (**F**) The proliferation of the CD4^+^ T lymphocytes isolated from the splenocytes was analyzed by flow cytometric microscopy after activation in vitro with an HBsAg peptide. (**G**) The proliferation of the CD8^+^ T lymphocytes isolated from the splenocytes was analyzed by flow cytometric microscopy after activation in vitro with an HBsAg peptide. ((**a**) PBS; (**b**) GMI-HBVac; (**c**) CTX; (**d**) CTX+GMI-HBVac; (**e**) MAF; (**f**) MAF + GMI-HBVac). * (*p* < 0.05) indicates significant difference, ** (*p* < 0.01) indicates highly significant difference, and *** (*p* < 0.001) indicates highly significant difference.

**Figure 6 vaccines-11-01026-f006:**
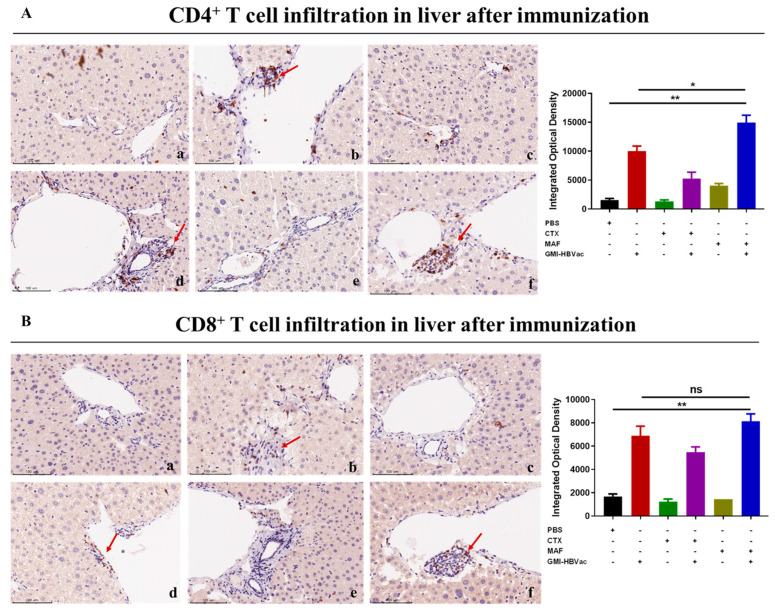
Immunization with MAF + GMI-HBVac leads to an increased number of T cells populating the livers of mice that have been infected with rAAV8-1.3HBV. (**A**) Immunohistochemical analysis of liver tissue was carried out with specific anti-CD4 antibodies used to stain the tissue (red arrow). (**B**) Immunohistochemical analysis of liver tissue was carried out with specific anti-CD8 antibodies used to stain the tissue (red arrow). ((**a**) PBS; (**b**) GMI-HBVac; (**c**) CTX; (**d**) CTX + GMI-HBVac; (**e**) MAF; (**f**) MAF + GMI-HBVac. ns (*p* > 0.05) indicates non-significant difference, * (*p* < 0.05) indicates significant difference, ** (*p* < 0.01) indicates highly significant difference.

**Figure 7 vaccines-11-01026-f007:**
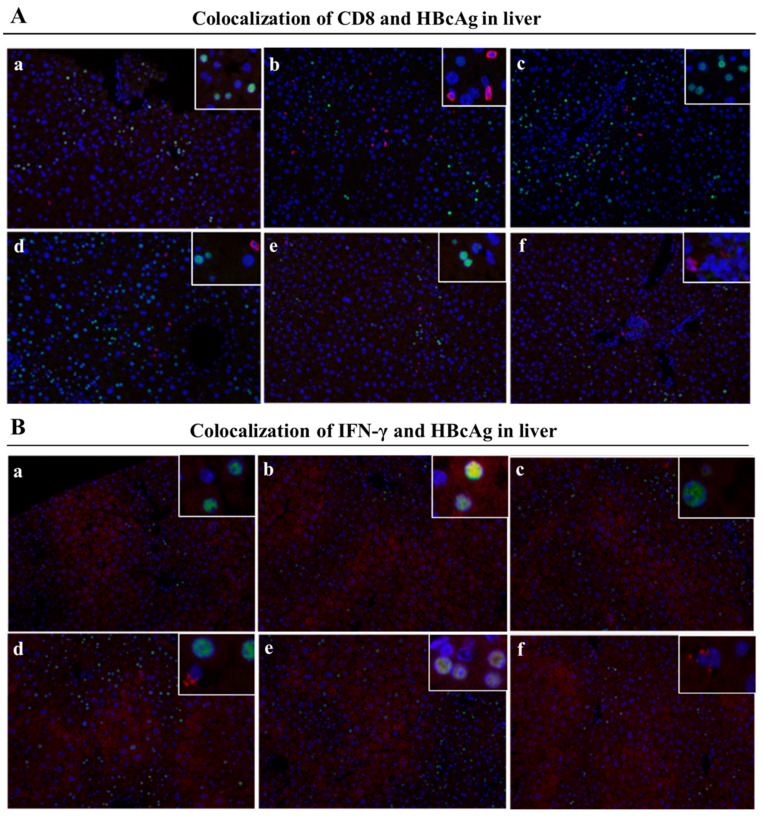
Immunization with MAF + GMI-HBVac boosts the immune response in the livers of mice that have been infected with rAAV8-1.3HBV. (**A**) Liver tissue sections were stained with immunofluorescent labels: (**A**) CD8 (red) and HBcAg (green); (**B**) IFN-γ (red) and HBcAg (green). (**a**) PBS; (**b**) GMI-HBVac; (**c**) CTX; (**d**) CTX + GMI-HBVac; (**e**) MAF; (**f**) MAF + GMI-HBVac.

**Figure 8 vaccines-11-01026-f008:**
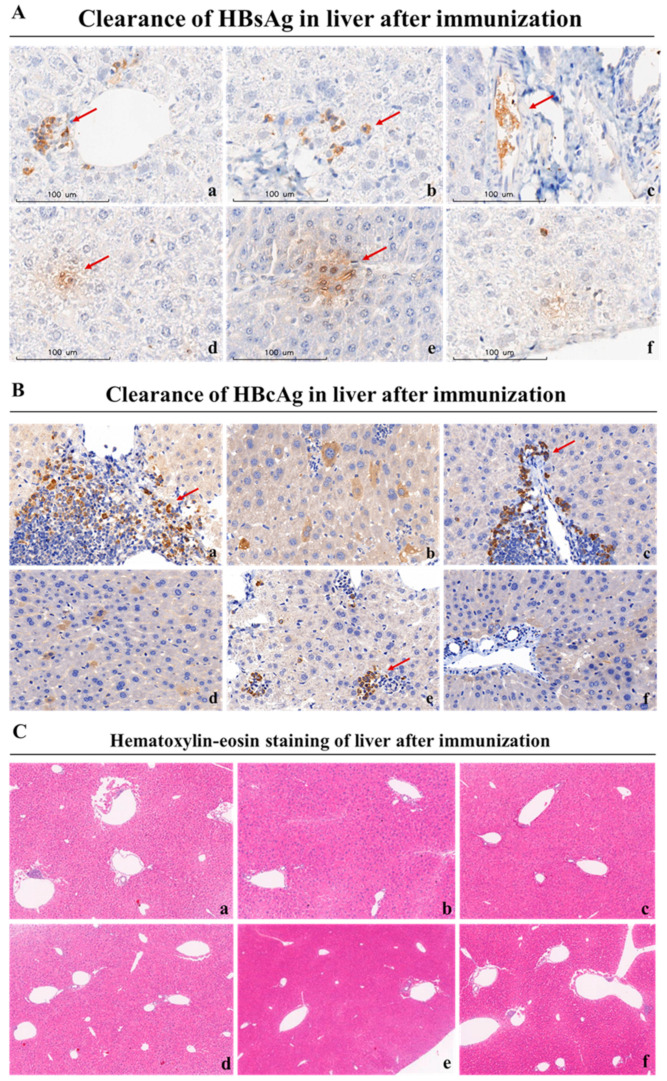
Liver HBV antigen clearance in rAAV8-1.3HBV-infected mice after treatment. (**A**) Immunohistochemistry with anti-HBsAg (red arrow). (**B**) Immunohistochemistry with anti-HBcAg (red arrow). (**C**) H&E. stain. (**a**) PBS; (**b**) GMI-HBVac; (**c**) CTX; (**d**) CTX + GMI-HBVac; (**e**) MAF; (**f**) MAF + GMI-HBVac.

**Figure 9 vaccines-11-01026-f009:**
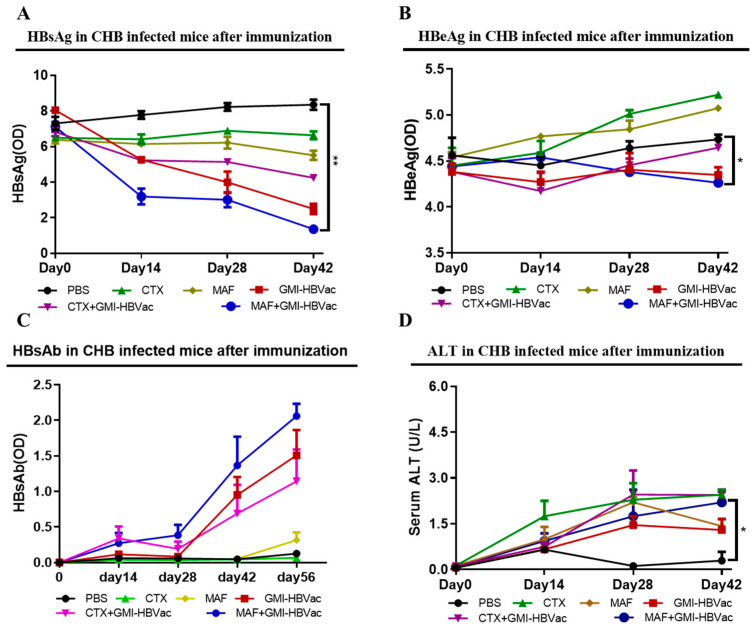
The administration of the MAF + GMI-HBVac regimen results in HBV antigen seroclearance in rAAV8-1.3HBV-infected animals. The serum levels of HBsAg (**A**), HBeAg (**B**), and HBsAb (**C**) were determined by ELISA. (**D**) Serum ALT levels. * (*p* < 0.05) indicates significant difference, ** (*p* < 0.01) indicates highly significant difference.

**Table 1 vaccines-11-01026-t001:** Regimens for immunization.

Group	Dose
PBS	150 μL
GM-CSF + IFN-α + rHBVvac (GMI-HBVac)	10 μg + 10,000 IU + 1 μg
CTX	15 mg/kg
CTX + GMI-HBVac	15 mg/kg + 10 μg + 10,000 IU + 1 μg
MAF	2 μg/mL
MAF + GMI-HBVac	2 μg/mL + 10 μg + 10,000 IU + 1 μg

## Data Availability

Data sharing will be available on request.

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
