# Peer review of "Mafosfamide Boosts GMI-HBVac against HBV via Treg Depletion in HBV-Infected Mice"

_vaccines, 2023, doi:10.3390/vaccines11061026_

Round 1

Reviewer 1 Report

In their present manuscript, the authors assessed a combination strategy using Mafosfamide pre-treatment to improve therapeutic vaccine anti-HBV effect. The idea of modulating immune response affected by HBV infection combined with induction of specific anti-HBV response using a therapeutic vaccine is interesting.

However, in the present manuscript I noticed several problem that I would like to be clarified by the authors.

First, maybe it is not a mistake from the authors, but some supplementary data are mentioned in the manuscript and I do not have access to it for the review. Some of these supplementary figures are mentioned in the text but it starts to number 2. Other are number 4 and 5. Why supplementary fig 1 and 3 mentioned in the text ? If those figure exist.

One of my major concern is about the methods used. Here are not described the isolation methods from organs (APC, splenocytes...).

The purity of the drugs used is not mentioned. Several materials are written without their references (e.g. IFN-a no reference and no subtype of IFN-a as IFN-a2). The authors used the PBS as a control for the treatment of mice but are the different compounds and vaccines diluted in PBS as well ?

For flow cytometry, the gating strategy should be explained and shown (maybe in supplementary) as well as the method used for extracellular and intracellular staining.

qPCR is mentioned with a limit of detection of 100 UI/ml but no result is given in proper DNA quantification (only delta Ct).

fig 1 result part 1:

Percentage of infected hepatocytes with HBs/HBc positive staining should be mentioned.

some mice shown a very low level of antigen but the author did not mentioned it in the text.

fig 1D the results of HBV DNA is mentioned in the legend and the text but here is HBsAb instead. Why the number of sera analyzed for ALAT is different from the other parameters (5 control and 13 HBV instead of 23 HBV in the 3 other graphs.)

"Peripheral blood monocytes" line 262 is peripheral blood mononuclear cells for PBMC ?

fig 2 and 3.

Data from CTX concentration are missing in fig 2 and figure 3. Why the doses used for CTX are so much higher than for the MAF ?

All the compounds have been tested in infected mice, but is the T reg depletion effect the same in healthy treated mice ? Is there any cytotoxicity in general and on the liver by CTX and MAF treatment ?

Is the decreasing effect of T reg population affecting other population (whole PBMC number ?). The cell percentage in fig 3B is calculated on total T cells or PBMCs ?

Number of mice for each group is not mentioned here. Why the trend of mouse weight and weight change looks so different ?

Fig 4: DC stained with CD11c and CD11b represent only a specific population of DC and not the whole population of these cells. Some other marker for the activation should have been assessed than the CD80 alone.

Fig 5: Is there any significativity with the PBS group in Fig 5B ? All other parameters are significant but not this one ?

Fig 6: Inflation has not reach the liver yet. Please correct for "infiltration"

Fig 7: name of the group is not in the legend. Is this colocalization representative of stronger inflammation ? Could it be analyzed more precisely using an image software analysis ?

Fig 8: Same remark as fig 7 please quantify the positive signal to have a better comparison with an image software.

Fig 9: How to explain the significant decrease of the HBV antigens whereas the HBV DNA is not ? In patients HBV antigens and mainly HBsAg is the most difficult parameter to affect using treatment. How the authors can described an active cccDNA level without assessing this parameter in ligne 543 for fig 9 ?

CTX and MAF seem to have a cytotoxic effect on the liver according to fig 9E. Please comment it.

The authors mentioned a potential effect on the MDSC on the immune response against HBV. Did they evaluate the presence of those cells in the liver tissue and if MAF treatment could affect them ?

Fig 10: A conclusion figure is always appreciated. Could you please complete the legend for it ?

Reviewer 2 Report

Manuscript entitled “Mafosfamide combine with GMI-HBVac against HBV via Treg depletion in HBV-infected mice” showed quality work. Manuscript has updated data but the major weakness is the discussion portion.

Introduction Line 41. “As of 2019, 2.3 billion people were infected with HBV worldwide”. This data is not correct.

Please cite leading research groups work on HBV published in Lancet group including

https://www.sciencedirect.com/science/article/pii/S2468125322001248

https://www.sciencedirect.com/science/article/abs/pii/S2468125318300566

Line 43-45. “In Asia, the percentages of chronic HBV infection in adults range between 5% and 10%, posing a serious public health concern.” this data is not correct. Use above papers to get authentic data of HBV in Asia.

In introduction, add a paragraph about the available HBV drugs in the market, and how much effective these drugs are?

Add paragraph about the WHO viral hepatitis strategy and progress and targets for HBV diagnostics and treatment.

https://www.ncbi.nlm.nih.gov/pmc/articles/PMC6262254/

In methodology, symbol for microgram is not correct.

Results shows extensive work.

Discussion part is not well written. Results are not discussed with the already published data. Try to get latest references and discuss your results.

References are old. Try to cite 70% of references from the last 5 years.

Round 2

Reviewer 1 Report

Dear authors thank you for the manuscript edition. Please check the text one more time as errors have been added with the corrections.

Author Response

Dear authors thank you for the manuscript edition. Please check the text one more time as errors have been added with the corrections.

Response: Thanks for your suggestion. We have asked Dr. Douglas Lowrie, who is a retired immunologist, proofread our entire manuscript with several correction s and polishings.